ecology/physiology/biochemistry

protein turnover, slow growth, temperature limitation, cryobiology, Antarctic, protein synthesis

**Author for correspondence:**
Keiron P. P. Fraser
e-mail: keiron.fraser@plymouth.ac.uk

# Life in the freezer: protein metabolism in Antarctic fish

Keiron P. P. Fraser[1], Lloyd S. Peck[2], Melody S. Clark[2], Andrew Clarke[2] and Simeon L. Hill[2]

[1]Marine Station, University of Plymouth, Artillery Place, Coxside, Plymouth PL4 OLU, UK
[2]British Antarctic Survey, Natural Environment Research Council, High Cross, Madingley Road, Cambridge CB3 OET, UK

KPPF, 0000-0001-5491-8376; SLH, 0000-0003-1441-8769

Whole-animal, *in vivo* protein metabolism rates have been reported in temperate and tropical, but not Antarctic fish. Growth in Antarctic species is generally slower than lower latitude species. Protein metabolism data for Antarctic invertebrates show low rates of protein synthesis and unusually high rates of protein degradation. Additionally, in Antarctic fish, increasing evidence suggests a lower frequency of successful folding of nascent proteins and reduced protein stability. This study reports the first whole-animal protein metabolism data for an Antarctic fish. Groups of Antarctic, *Harpagifer antarcticus*, and temperate, *Lipophrys pholis*, fish were acclimatized to a range of overlapping water temperatures and food consumption, whole-animal growth and protein metabolism measured. The rates of protein synthesis and growth in Antarctic, but not temperate fish, were relatively insensitive to temperature and were significantly lower in *H. antarcticus* at 3°C than in *L. pholis*. Protein degradation was independent of temperature in *H. antarcticus* and not significantly different to *L. pholis* at 3°C, while protein synthesis retention efficiency was significantly higher in *L. pholis* than *H. antarcticus* at 3°C. These results suggest Antarctic fish degrade a significantly larger proportion of synthesized protein than temperate fish, with fundamental energetic implications for growth at low temperatures.

## 1. Introduction

The Southern Ocean is characterized by low, stable temperatures and strong seasonal variation in sea ice, light and primary productivity [1,2]. Resting or standard metabolic rates of Antarctic ectotherms are generally considered low [3,4], and growth rates reduced [5–8] in comparison to temperate and tropical species. Recent research has suggested that reduced growth rates in Antarctic organisms may be driven by reduced rates of protein synthesis and/or protein retention [4,9,10]. There are, however, a

few Antarctic invertebrate species in which comparatively high growth rates have been reported [11–13]. Although, even in these cases, growth rates are significantly slower than the fastest growth rates for temperate and tropical relatives [4].

Growth is fundamental to the life history of an organism and is primarily the result of the synthesis and retention of proteins, an energetically expensive process accounting for a significant proportion of an organism's total energy budget [9,14–16]. Although much work has gone into examining protein synthesis in a wide range of marine organisms [3,7,16–19], few studies have examined both protein synthesis and growth in the same animals (for example [7,18,20–24]) or protein synthesis in ecologically similar species from widely differing latitudes [17,19]. Thus, the relationship between growth and protein synthesis and how it is affected by long-term environmental conditions are poorly understood. This situation is confounded by a lack of information on whole-animal protein synthesis, degradation and growth rates, i.e. protein metabolism in Antarctic vertebrates, although some tissue-specific protein synthesis rates for fish have been published [25–27].

Previous studies of protein metabolism in the Antarctic limpet, *Nacella concinna*, have suggested that whole-animal protein metabolism in this species is very inefficient in comparison with temperate and tropical species [7]. Studies in Antarctic notothenioid fishes have additionally demonstrated very high tissue levels of ubiquitinated proteins, an indicator of denatured proteins [28,29]. Most Antarctic invertebrates examined to date also exhibit high whole-animal or tissue RNA to protein ratios, suggesting that the translational machinery is less efficient at Antarctic water temperatures (e.g. [3,7,8,30,31] but also see [17]).

Additionally, Antarctic fish species, along with all other Antarctic marine invertebrates studied, maintain high constitutive levels of members of the 70 kDa heat shock protein (HSP 70) family, which are critical in ensuring the correct folding of nascent polypeptides [32]. In Antarctic fish, this high constitutive level of expression is achieved by an alteration in the promotor region of the inducible form of HSP 70 [33].

Taken together, these studies suggest that there may be specific challenges to synthesizing and maintaining functional proteins at polar water temperatures [4,10,34,35]. Further *in vivo* data are clearly required to improve understanding of the efficiency of synthesizing proteins at these low water temperatures. Here, we investigate the effect of evolutionary adaptation to habitat on fish protein metabolism, through a comparative study of protein synthesis, retention and RNA to protein ratios in two ecologically similar fish, one from Antarctica and the other from temperate waters. Whole-animal rates of protein synthesis were measured rather than just white muscle, so that rates of protein degradation could be estimated and directly compared with food consumption and whole-animal growth rates. The hypothesis being tested in this study is that evolutionary adaptation to low Antarctic water temperatures will result in low levels of whole-animal protein synthesis and high rates of protein degradation.

# 2. Material and methods

## 2.1. Collection and husbandry of fish

The Antarctic spiny plunderfish, *Harpagifer antarcticus*, were collected by hand from sublittoral sites at Rothera Point, Adelaide Island, Antarctic Peninsula (67°34′07″ S, 68°07′30″ W) during the austral summer by SCUBA divers and held in a through-flow aquarium until return to the UK in a refrigerated transport system. In the UK, fish were maintained in a recirculating flow aquarium, under a 12 h light : dark regime (water temperature 0°C ± 1.0°C, salinity 34–36), for at least one month before experimental temperature acclimatization started. The fish were fed twice weekly to satiation on shelled krill (*Euphausia superba*). The shanny, *Lipophyrs pholis*, were collected subtidally in Weymouth (UK) by a commercial supplier and fed on chopped white fish twice weekly (New Zealand Hoki, *Macruronus novaezelandiae*) as they refused krill. The mean water temperature of the *L. pholis* stock tank was 14°C ± 1.0°C, salinity 34–36 and a 12 h light : dark regime. All fish were held and experimental work carried out at the British Antarctic Survey, Cambridge, UK. Experimental work was carried out in July and December with *H. antarcticus* and February to May with *L. pholis*, effectively winter to spring for each species in its respective natural habitat. The species selected for the study were ecologically similar; both are crypto-benthic, feed on small invertebrates, are found intertidally or in very shallow water and are of a similar maximum body length. *Harpagifer antarcticus* is only found in Antarctica, and at Rothera will have experienced water temperatures ranging between −1.89°C and approximately 2°C [2]. *Lipophyrs pholis* is distributed from southern Norway to Morocco

and inhabits a range of water temperatures exceeding those used in this study [36]. In both species, animals used in experimental work were adult.

## 2.2. Temperature acclimatization

Experimental animals were weighed in water (±0.1 g) and digitally photographed to allow individual identification. The mean mass of *H. antarcticus* was $13.76 \pm 0.49$ g and *L. pholis* $16.25 \pm 0.82$ g. The fish were placed in pairs, in jacketed aquaria ($18.8 \times 20.3 \times 22.0$ cm) containing aerated seawater at their stock tank temperature. Water temperatures were maintained ($\pm 0.1°C$) using a thermocirculator (Grant Instruments, Cambridge, UK). Once each day, the aquaria were divided in half using a Perspex insert, thereby allowing fish to be fed individually to satiation and their food consumption measured (± 0.1 g).

Adjustment of aquaria to the required experimental water temperatures (*H. antarcticus*, −1°C (number of fish per temperature treatment, $n = 8$), 1°C ($n = 8$) and 3°C ($n = 16$), *L. pholis*, 3°C ($n = 15$), 8°C ($n = 8$), 13°C ($n = 8$) and 18°C ($n = 12$)) was carried out at a rate of $0.1°C\ h^{-1}$ (*H. antarcticus*) or $0.5°C\ h^{-1}$ (*L. pholis*), up to a maximum of $0.8°C\ d^{-1}$ (*H. antarcticus*) or $4°C\ d^{-1}$ (*L. pholis*). Fish were weighed once the experimental temperature was reached, and fish were acclimatized to the experimental temperature for 28 d prior to the measurement of protein synthesis, as Antarctic fish can take three to four weeks to acclimatize to elevated temperatures and are the slower of the two species to acclimatize [37]. The fish were weighed on the first and penultimate day of the 28 d acclimatization at the experimental temperature. Specific growth rates (SGRs) were calculated for each fish using the following equation [38]:

$$SGR = \frac{\ln(M_2) - \ln(M_1)}{\Delta t} \times 100,$$

where SGR is expressed as % body mass $d^{-1}$, $M_1$ and $M_2$ represent the mass at the start and end of the acclimatization period, respectively and $\Delta t$ is the time in days between mass measurements.

## 2.3. Protein synthesis measurement

Whole-animal protein synthesis rates were measured using a modification of the flooding dose method [7,39,40]. Fish were not fed on the day that protein synthesis was measured but had been fed on the previous day. Groups of fish were injected in the peritoneum with a flooding dose of unlabelled and $^3H$ labelled phenylalanine ($10\ \mu l\ g^{-1}$ fish wet mass of 135 mmol $l^{-1}$ L-(2,6–$^3H$) phenylalanine at 3.6 MBq $ml^{-1}$ (GE Healthcare, Little Chalfont, UK)). Fish were killed and frozen in liquid nitrogen after 1, 2 and 4 h to allow validation of the flooding dose time-course (*H. antarcticus*, 3°C; *L. pholis*, 3°C and 18°C). In other experimental treatments, fish were killed after 2 h (*H. antarcticus*, −1°C and 1°C; *L. pholis*, 8°C and 13°C). Fish were killed by a sharp blow to the head and subsequent destruction of the brain. Baseline, pre-injection phenylalanine concentrations were measured in both species ($n = 10$) to allow the calculation of phenylalanine flooding levels. All samples were frozen in liquid nitrogen and stored at −80°C prior to analysis.

## 2.4. Protein synthesis and RNA content: sample analysis

The frozen fish were homogenized in a known volume of ice-cold, 0.2 M perchloric acid (PCA) at a ratio of 2 ml of PCA per 100 mg of tissue. The homogenate was mixed thoroughly, and a 2 ml aliquot removed for analysis. The aliquot was centrifuged (Eppendorf 5810R, swing bucket rotor, 3980*g*, 10 min, 4°C) to separate the protein precipitate and RNA from the intracellular free pool [41]. The pellet was washed twice with 0.2 M PCA, which has been found to be sufficient to remove unbound labelled phenylalanine [41]. The supernatant was decanted, and the NaOH soluble protein in the pellet was measured using bovine serum albumin (Sigma-Aldrich, Poole, UK) as the standard [42]. Total RNA was measured by comparing the sample concentrations with known RNA standard (Type IV, calf liver, Sigma) concentrations, determined spectrophotomically at 665 nm after reaction with an acidified orcinol reagent [43]. Subsequently, the pellet was washed twice with 0.2 M PCA and hydrolysed in 6 M HCl for 18 h at 110°C. The acid was removed from the hydrolysed protein residue using repeated washes of distilled water and rotary evaporation to dryness between washes, before the residue was re-suspended in 0.5 M sodium citrate buffer (pH 6.3). The phenylalanine concentration of the hydrolysed protein residue, injection solution and the intracellular free-pools was measured fluorometrically, after enzymatic conversion of the phenylalanine to β-phenylethylamine [7,41]. The specific radioactivities of the free-pools, protein and injection solutions were measured using

scintillation counting (Wallac 1409 LSC, Packard Bioscience Hionic-Fluor scintillant, 34% $^3$H counting efficiency) and expressed as disintegrations per minute (d.p.m.) nmol$^{-1}$ phenylalanine.

The fractional rate of protein synthesis ($k_s$) was calculated from

$$k_s = \frac{S_b}{S_a} \times \frac{100}{t} \times 1440,$$

where $k_s$ = % protein mass synthesized per day (% d$^{-1}$), $S_b$ = specific radioactivity of protein-incorporated radiolabel (d.p.m. nmol$^{-1}$ phe), $S_a$ = specific radioactivity of the intracellular free-pool (d.p.m. nmol$^{-1}$ phe), $t$ = incorporation time from injection of radiolabel to death in minutes and 1440, the number of minutes in a day [39]. Validation of the flooding dose procedure used is described in the electronic supplementary material, figure S1. Protein growth rates ($k_g$, % d$^{-1}$) were calculated using the SGR equation, but with $M_1$ and $M_2$ representing the protein mass at the beginning and end of the growth period, respectively. Protein synthesis retention efficiency (PSRE) is the fraction of synthesized protein that is retained in new tissue and is a key factor in determining the growth efficiency of an organism [41]. PSRE was calculated as

$$\text{PSRE} = \frac{k_g}{k_s} \times 100.$$

## 2.5. Calculation of RNA translational efficiencies

Tissue RNA to protein ratios were expressed as µg RNA mg$^{-1}$ protein. The translational efficiency of the RNA ($k_{RNA}$, mg protein mg$^{-1}$ RNA d$^{-1}$) was calculated using the following equation [44]:

$$k_{RNA} = \frac{10 \times k_s}{\text{RNA to protein ratio}}.$$

## 2.6. Statistical analysis

All data are expressed as means ± standard error (SEM). Statistical analysis was carried out using Minitab version 19 (Minitab, Coventry, UK). Prior to statistical analysis, data were checked for normality and homogeneity of variances using the Anderson–Darling and Levene's tests [45]. Parametric data were analysed using analysis of variance (ANOVA) and Tukey honestly significant difference (HSD) tests, while non-parametric data were analysed using the Kruskal–Wallis test. Student's $t$-test was used to assess differences in metrics between species. Two-tailed significance was accepted at $p < 0.05$.

# 3. Results

There were no significant differences in fish masses between temperature groups or species; therefore, data were not mass standardized prior to analysis (see electronic supplementary material, table S1).

## 3.1. Specific growth rate and ration

There was a small, but non-significant increase in *H. antarcticus* SGR with temperature, possibly because of the narrow range of temperatures used (figure 1*a*, regression analysis, $p = 0.09$, $F = 3.13$). By contrast, there was a quadratic relationship between SGR and temperature in *L. pholis* (figure 1*a* and table 1). At the common temperature of 3°C, *H. antarcticus* SGR was significantly lower (49%) than *L. pholis* (figure 1*a* and table 2).

Food consumption increased linearly with temperature in both species (figure 1*b* and table 1) with no species-specific difference in slope (figure 1*b*, $F = 3.5$, $p = 0.065$). At 3°C, *H. antarcticus* consumed significantly less (20%) food than *L. pholis* (figure 1*b* and table 2).

## 3.2. Validation of the flooding dose methodology

As protein synthesis has not been previously measured in either of the study species, full validation of the flooding dose methodology was considered essential, using accepted criteria [34]. Intracellular free-pool-specific radioactivities increased rapidly and were elevated and stable over the course of the experiment. The incorporation of radiolabelled phenylalanine into tissue protein was significant and linear, with an intercept not significantly different from zero, indicating incorporation of the radiolabelled phenylalanine occurred rapidly after injection (see electronic supplementary material, figure S1 and table S2).

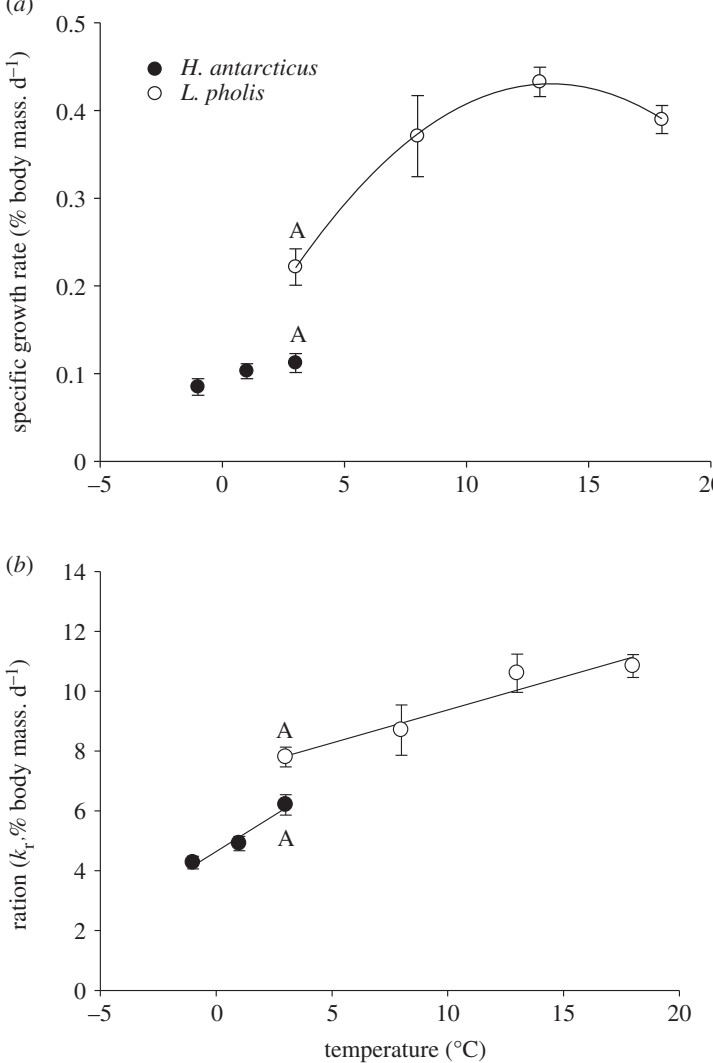

**Figure 1.** Specific growth rates (SGRs, *a*) and ration ($k_r$, *b*) in groups of *H. antarcticus* (●) acclimatized to −1 (8), +1 (8) and 3℃ (16) and *L. pholis* (○) acclimatized to +3 (15), 8 (8), 13 (8) and 18℃ (12) for 28 d. Numbers in parentheses are *n* numbers for each data point. Overlapping treatments at 3℃ are significantly different between species if marked with A. All data points are mean ± SEM. For regression equations, see table 1.

### 3.3. Fractional protein synthesis, growth and degradation

The data for growth rate and ration point to a lower growth efficiency in the Antarctic species. Fractional protein synthesis, growth and degradation rates were independent of temperatures between −1°C and 3°C in *H. antarcticus* (figure 2*a–c*; ANOVA, $k_s$, $p = 0.091$, $F = 3.06$; $k_g$, $p = 0.087$, $F = 3.14$; $k_d$, $p = 0.673$, $F = 0.4$). However, there was a small non-significant increase in protein synthesis and protein growth with temperature. In *L. pholis*, the relationship between fractional protein synthesis and temperature was best described by a linear model, protein growth and temperature; a quadratic model, protein degradation and temperature; and a cubic model (figure 2*a–c* and table 1). At the overlapping temperature of 3°C, the fractional rate of protein synthesis and protein growth (figure 2*a,b* and table 2) was significantly lower in *H. antarcticus* than *L. pholis*, but there was no significant difference in protein degradation rates between the species, except when *L. pholis* were held at 18°C (figure 2*c* and table 2).

### 3.4. Protein synthesis retention efficiency

PSRE was independent of temperature between −1°C and 3°C in *H. antarcticus* (figure 2*d*, ANOVA, $p = 0.525$, $F = 0.41$), whereas in *L. pholis*, there was a cubic relationship (figure 2*d* and table 1). The PSRE was significantly lower (37%), at 3°C, in *H. antarcticus* than *L. pholis* (figure 2*d* and table 2).

**Table 1.** Regression equations for significant ($p < 0.05$) relationships between temperature and specific growth rate, food consumption, protein synthesis, protein growth, protein degradation and protein synthesis retention efficiency.

| species | metric | $r^2$ | equation |
|---|---|---|---|
| *L. pholis* | SGR | 52.5 | $y = 0.084 + 0.052x - 0.002x^2$ |
| *H. antarcticus* | $k_r$ | 37.8 | $y = 4.667 + 0.502x$ |
| *L. pholis* | $k_r$ | 38.9 | $y = 7.218 + 0.213x$ |
| *L. pholis* | $k_s$ | 54.8 | $y = 0.278 + 0.039x$ |
| *L. pholis* | $k_g$ | 52.4 | $y = 0.118 + 0.062x - 0.002x^2$ |
| *L. pholis* | $k_d$ | 51.1 | $y = 0.145 + 0.121x - 0.015x^2 + 0.001x^3$ |
| *L. pholis* | PSRE | 44.9 | $y = 87.35 - 9.66x + 1.482x^2 - 0.059x^3$ |
| *H. antarcticus* | RNA: protein | 25.3 | $y = 23.84 - 1.448x$ |
| *L. pholis* | RNA to protein | 42.9 | $y = 12.56 - 0.215x$ |
| *H. antarcticus* | $k_{RNA}$ | 21.7 | $y = 0.097 + 0.011x$ |
| *L. pholis* | $k_{RNA}$ | 70.5 | $y = 0.181 + 0.050x$ |

**Table 2.** Student's *t*-test results for the comparison of metrics measured in *Harpagifer antarcticus* and *Lipophrys pholis* at 3°C.

| metric measured | T | p |
|---|---|---|
| specific growth rate | −4.67 | <0.001 |
| food consumption | −3.39 | <0.01 |
| protein synthesis | −7.37 | <0.001 |
| protein growth | −6.21 | <0.001 |
| protein degradation | 1.94 | NS |
| protein synthesis retention efficiency | −4.81 | <0.001 |
| RNA to protein | 8.72 | <0.001 |
| RNA translational efficiency | −10.62 | <0.001 |

## 3.5. RNA to protein ratios and RNA translational efficiencies

The relationship between the RNA to protein ratio and temperature was linear, with a negative and significantly different ($p < 0.01$, $F = 15.83$) slope for both *H. antarcticus* and *L. pholis* (figure 2*e*). The RNA to protein ratio was significantly higher (55%) in *H. antarcticus* than *L. pholis* at 3°C (table 2).

The relationship between RNA translational efficiency ($k_{RNA}$) and temperature was linear for both *H. antarcticus* and *L. pholis* (figure 2*f*); however, the slopes of the regression lines were significantly different ($p < 0.05$, $F = 5.53$). $k_{RNA}$ was significantly (59%) lower in *H. antarcticus* than in *L. pholis* at 3°C (table 2).

## 4. Discussion

Soft tissue growth rates in organisms are largely determined by the rate and efficiency with which proteins are synthesized and retained [34]. The energetic cost of protein metabolism dominates the overall energetic cost of macromolecular synthesis and the resultant chemical energy retained within new tissue [34,46]. It is well established that temperature and food consumption are the main drivers of protein synthesis rates and in turn growth in fish [47–51]. However, understanding of protein metabolism in Antarctic fish is currently poor. This study examined the effects of biologically relevant temperature ranges on the rate and efficiency of protein metabolism in an Antarctic and a temperate teleost. In doing so, we aimed to understand the effect of evolutionary adaptation to habitat, on fish protein metabolism.

Most studies of the effects of ambient water temperature on protein metabolism have concentrated on the effects of either two or a very limited range of temperatures on protein synthesis in a single fish

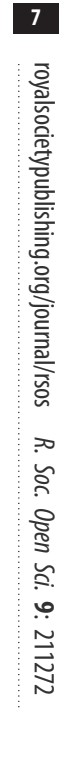

**Figure 2.** The fractional rate of protein synthesis ($k_s$, a), protein growth ($k_g$, b), protein degradation ($k_d$, c), protein synthesis retention efficiency (PSRE, d), RNA to protein ratio (e) and RNA translational efficiency ($k_{RNA}$, f) in groups of *H. antarcticus* (●) acclimatized to −1 (8), +1 (8) and 3°C (16) and *L. pholis* (○) acclimatized to +3 (15), 8 (8), 13 (8) and 18°C (12) for 28 d. Numbers in parentheses are *n* numbers for each data point. Overlapping treatments at 3°C are significantly different between species if marked with A. All data points are mean ± SEM. For regression equations, see table 1.

species [24,48,52–54], although recently protein synthesis rates have been reported in several amphipod species across a wide natural thermal gradient [19]. Very few studies have examined protein synthesis, degradation and growth simultaneously in a fish species or multiple species, over a range of temperatures [24,48,55].

## 4.1. The effect of temperature on ration

The effect of temperature on whole-animal fractional protein synthesis rate ($k_s$, % d$^{-1}$) is complicated by the influence of temperature on food consumption. If a fixed or restricted ration is fed, then no temperature-related increase in whole-animal protein synthesis may occur [56,57]; however, if animals are fed to satiation, rates of protein synthesis will increase up to a species-specific thermal optimum before declining (e.g. [24,31,58,59]).

In the present work, both species were fed to satiation and food consumption increased linearly with increasing temperature, at a similar rate in both fish species. There was no significant difference in respective regression slopes (figure 1b); however, at 3°C, L. pholis consumed a 20% larger ration than H. antarcticus. Harpagifer antarcticus have a limited ability to increase their aerobic factorial scope after feeding in comparison with lower latitude fish species, and this has been interpreted as a fundamental post-prandial restriction in the ability to upregulate one or more significant cellular processes during the specific dynamic action (SDA) response [60]. Protein synthesis, degradation and growth form a significant proportion of the SDA response [31,61]. It is therefore possible that the lower $k_r$ and $k_s$ observed in H. antarcticus at 3°C, in comparison with L. pholis may be due to a limited physiological scope to process food or absorb nutrients and thereby use the products of digestion for protein metabolism during the SDA response. Such limited capacities to raise metabolic rates during the SDA response have been widely reported in Antarctic marine species [62].

## 4.2. The effect of temperature on protein synthesis

In H. antarcticus, $k_s$ did not significantly increase with temperature, although there was a small non-significant trend; in contrast, an increase in acclimatization temperature caused a significant increase in L. pholis $k_s$ (figure 2a). The lack of a significant thermal response in H. antarcticus may be the result of the narrow range of water temperatures this species has evolved within and was acclimatized to in this study. However, it is worth noting that H. antarcticus were exposed to a 4°C (−1°C–3°C) temperature range without eliciting a significant change in $k_s$, $k_g$ and $k_d$, while a 5°C increase (3°C–8°C) in water temperature had a large effect on $k_s$ and $k_g$ in L. pholis, suggesting a lower thermal sensitivity of protein metabolism in the Antarctic species. In the Antarctic fish, Pagothenia borchgrevinki held at −1°C, acute exposure to 4°C resulted in basal metabolic rates increasing by 41% [63]. However, if the same species were acclimatized to 4°C for 28 d, resting metabolic rates were not significantly different to those fish held at −1°C [63]. Previous thermal history is therefore important in determining physiological responses and associated Q10 values.

The whole-animal fractional rates of protein synthesis in L. pholis (0.39–0.99% d$^{-1}$) fall within the range previously reported for other ectotherm species (for reviews, see [34,64]); however, the results are at the lower end of the reported spectrum. By contrast, whole-animal $k_s$ in H. antarcticus (0.21–0.25% d$^{-1}$) was lower than reported in any teleost species at their typical habitat temperature, although whole fish $k_s$ has not previously been reported in any polar teleost. Data for whole-body $k_s$ in adult Antarctic invertebrates are limited to a few studies. In the Antarctic limpet, Nacella concinna, $k_s$ ranged between 0.27 and 0.8% d$^{-1}$ depending on season, with the authors suggesting that the change in $k_s$ was due to seasonal changes in food consumption, not the 2°C change in water temperature [7,8,9]. In the Antarctic isopod, Glyptonotus antarcticus, protein synthesis rates of 0.24% d$^{-1}$ have been measured [59], while in the Arctic whole-animal protein synthesis, rates have been reported in two species of marine gammarid amphipods: 0.25 and 1.25% d$^{-1}$ in Gammarus oceanicus and G. setosus respectively, at 5°C [19]. The low whole-animal $k_s$ rates for H. antarcticus reported here therefore seem to be more in line with reported data from Antarctic marine invertebrates, than previously reported non-polar fish data.

Evidence of temperature-related, seasonal depression of protein synthesis in a range of tissues has been reported in the cunner, Tautogolabrus adspersus a species found in the Northwest Atlantic and which exhibits inactivity at 4°C and cessation of feeding [65]. Seasonal depression of protein synthesis does not appear likely in the species used in the current study as both fed well at all temperatures (figure 1b) and both species are active and known to feed in their natural habitats year-round [66,67]. In the current study, the potential effects of seasonal changes in the light : dark regime on protein synthesis were removed by holding the animals and carrying out all experimental work at a standardized 12 : 12 h regime.

In the current study, whole-animal protein synthesis rates in the Antarctic species, H. antarcticus, were considerably lower than those measured in the temperate species, L. pholis, at water temperatures which the species typically inhabits. In turn, this provides further evidence that there is a fundamental thermal constraint on the synthesis of proteins in species that have evolved to live at very low temperatures [34,35]. Fractional protein synthesis rates ($k_s$) were also lower in H. antarcticus when compared with L. pholis at the same water temperature (figure 2a). The low rates of growth and embryonic development in Antarctic species reported elsewhere [4,10] may therefore be set not only by a limited and highly seasonal food supply but also by a reduced ability to synthesize functional proteins. Interestingly, a comparative in vitro study has demonstrated that, at least in eelpouts, the actual machinery of cellular protein synthesis does appear to be thermally compensated [17].

Previous studies in some fish species have demonstrated a degree of inter- and intra-specific phenotypic variation across latitudes, counteracting the effects of reduced temperature and growing season at higher latitudes and resulting in similar growth rates across a wide latitudinal range, a phenomenon termed countergradient variation [68,69]. However, in the current study, there was no evidence that growth rates in the Antarctic species examined could approach those seen in the temperate species, even when held at the same temperature, in spite of an increase in the RNA machinery required for protein synthesis.

By contrast, comparatively fast growth rates, when compared with most Antarctic species, have been reported in the bivalve *Adamussium colbecki* [70], echinoids [4] and bryozoans and spirobid worms [71], although it should be noted that the growth rates reported are still considerably lower than in temperate species held in optimum conditions.

## 4.3. The effect of temperature on protein growth

$k_g$ was not significantly affected by temperature in *H. antarcticus*, although a slight increase in rate was detectable. However, in *L. pholis*, $k_g$ increased significantly from 0.28% $d^{-1}$ at 3°C to a maximum of 0.54% $d^{-1}$ at 13.0°C, at which point further increases in temperature led to a reduction in the rate of $k_g$ to 0.49% $d^{-1}$ (figure 2*b*). The results for *L. pholis* are broadly similar to the limited whole-animal data available for other temperate species, such as the Atlantic wolfish, *Anarhichas lupus*, in that whole-animal $k_g$ increased up to an optimal temperature, before decreasing [24]. By contrast, in fed juvenile rainbow trout, whole-animal $k_g$ increased linearly with temperature between 5°C and 15°C, perhaps because the optimum temperature for protein growth was not exceeded [21].

Many studies have examined fish $k_g$ at only a single temperature [52–54]. The maximal rate of $k_g$ in *H. antarcticus* in this study was 0.11% $d^{-1}$, much lower than that reported in other teleost studies, for example 1.31% $d^{-1}$ in the Atlantic halibut (*Hippoglossus hippoglossus*) [20], 0.59% $d^{-1}$ in the rainbow trout (*Oncorhynchus mykiss*) [72] and 0.71% $d^{-1}$ in the flounder (*Pleuronectes flesus*) [54], similar values to those reported here for *L. pholis*. Therefore, protein growth rates reported here for *L. pholis* are similar to those reported previously in temperate fish, even when acclimatized to 3°C, whereas protein growth rates in *H. antarcticus* are considerably lower at all experimental temperatures.

## 4.4. The effect of temperature on protein degradation

By measuring protein growth over 28 d, and measuring the rate of protein synthesis at the end of this period, protein degradation was estimated as the difference between protein synthesis and protein growth [9,40,41,50,73]. This method relies on the assumption that the rate of protein synthesis measured at the end of the growth period is representative of protein synthesis across the entire growth period. Despite this constraint, this is currently the only method by which overall protein degradation rates can be measured *in vivo*, and hence, our understanding of protein degradation in animals is comparatively poor compared with protein synthesis. The complexity in measuring protein degradation is that the process occurs via several pathways, and therefore, fractional degradation rates cannot be directly calculated [74]. Methods have been developed that allow, for example, an insight into relative rates of protein degradation via one of the key protein degradation pathways, the ubiquitin-proteasome pathway [28,29]. However, the ubiquitin-proteasome pathway makes only a small contribution to protein degradation in fish muscle, a key tissue for understanding fish growth, with the majority of degradation occurring via the calpain and lysosomal-autophagic pathways [75]. Direct measurement of overall whole-animal fractional protein degradation rates in an organism would be a significant advance, but it is still currently not possible.

In the current study, $k_d$ was independent of temperature in *H. antarcticus* when measured between −1°C and +3°C and not significantly different between fish species when measured at the overlapping acclimatization temperature (figure 2*c*). Although *H. antarcticus* was acclimatized to 3°C, a temperature that is probably as high as the species will experience in its natural habitat and higher than at the site where specimens were collected for this study, no increase in $k_d$ was observed. In turn, this either suggests that protein degradation rates do not, or cannot, increase above the optimum growth temperature in *H. antarcticus* or that the animals in the study were not exposed to a temperature that exceeded the species-specific optimum growth temperature.

There is evidence that biochemical indicators of protein degradation are elevated in fish adapted to both cold temperature [76,77] and Antarctic water temperatures [28,29,33]. Elevated levels of ubiquitinated proteins in Antarctic fish suggest possible problems with protein folding and/or high

levels of protein degradation [28,29,78]. In the species in the current study, protein degradation was not significantly different between species at the same acclimatization temperature and was independent of temperature between −1°C and 3°C in *H. antarcticus*. Why $k_d$ does not decrease with temperature is currently unclear, but at low temperatures, a much larger proportion of the protein synthesized does appear to be immediately or rapidly degraded (figure 2*a,c,d*). Previous authors have suggested that the most likely explanation for high protein degradation rates at low temperatures is an inefficiency in protein folding after synthesis or potentially oxidative damage due to elevated tissue levels of oxygen free radicals [29]. In mammalian cells, 30% of newly synthesized proteins are defective and degraded within 10 min of synthesis [79]. Unfortunately, no similar studies have been carried out in ectotherms. It is also known that Antarctic invertebrates and fish do have elevated levels of antioxidant protection in comparison with lower latitude species [80,81] to counter increased formation of reactive oxygen species [82]. However, it is not currently clear whether this increased antioxidant capacity is adequate to maintain protein oxidation rates at similar levels to lower latitude fishes, although in eelpouts this does seem to be the case [83].

## 4.5. The effect of temperature on protein synthesis retention efficiency

PSRE is a measure of the proportion of synthesized protein which is retained as protein growth [41] and is fundamental in determining the overall growth of an organism [34]. PSREs have only previously been reported in a single Antarctic species, the limpet *Nacella concinna*, in which PSREs were over 50% lower than previously reported in any fish or mollusc [9]. PSREs reported for *H. antarcticus* in the current study are again at the lower end of those previously reported (for reviews see [34,35]), while at the optimum growth temperature of 13°C, *L. pholis* has one of the highest values reported (83.1%) and decreased significantly above this temperature (figure 2*d*).

PSRE was unaffected by temperature between −1°C and +3°C in *H. antarcticus* but was half that of *L. pholis* at 3°C. PSRE in *H. antarcticus* is low in comparison with *L. pholis*, because the protein synthesis rates in the Antarctic fish are low, while protein degradation rates are comparatively high when compared with the temperate species (figure 2*d*).

## 4.6. RNA to protein ratios and RNA translational efficiency

The changes in RNA to protein ratio and RNA translational efficiency ($k_{RNA}$) with water temperature in the current study fit the pattern seen in other ectotherm studies where a single species has been exposed to a range of temperatures [24,84] and in intra-specific comparisons [7,35], with RNA to protein ratios decreasing as water temperatures increase and $k_{RNA}$ increasing (figure 2*e,f*). However, even with a 55% higher RNA to protein ratio in *H. antarcticus* than *L. pholis* at 3°C, the Antarctic species still synthesized 59% less protein per unit of RNA. The energetic cost of RNA synthesis is not known in these species, but previous studies of combined RNA and DNA synthesis costs in hepatocytes from Antarctic fish have demonstrated they range between 24% and 35% of respiratory oxygen consumption [85]. If this is the case with *H. antarcticus*, then there must be a significant energetic cost of maintaining such high tissue RNA concentrations.

# 5. Concluding remarks

Our data provide the first protein metabolism data reported for any Antarctic fish species and add to an increasing body of evidence that synthesizing and retaining proteins at Antarctic water temperatures is problematic, in turn constraining maximum growth rate. We also demonstrate that in the Antarctic species, not only are the rates of growth and protein metabolism significantly lower than in the temperate species, even when held at the same water temperature, but the ability to increase the rates of these processes as water temperatures increase is considerably reduced. Additionally, this study provides one of the largest comparative studies of protein metabolism, growth and food consumption in fishes across a wide range of realistic habitat temperatures.

Ethics. All experimental work was approved by the British Antarctic Survey Ethical Review Committee and carried out under Home Office Project Licence PIL 80/8523.

Data accessibility. Data produced in this study are held by the Polar Data Centre and are available via the following link.

Fraser, K.P.P. (2021). Harpagifer antarcticus and Lipophrys pholis protein metabolism (v. 1.0) [Dataset]. NERC EDS UK Polar Data Centre: https://doi.org/10.5285/53737FBE-8CB8-4E3D-BC48-2B21E7C9D543.

Authors' contributions. K.P.P.F.: conceptualization, data curation, formal analysis, funding acquisition, investigation, methodology, project administration, resources, supervision, validation, writing—original draft, writing—review and editing; L.S.P.: conceptualization, funding acquisition, investigation, project administration, supervision, writing—original draft, writing—review and editing; A.C.: conceptualization, funding acquisition, investigation, project administration, supervision, visualization, writing—original draft, writing—review and editing; M.S.C.: project administration, resources, visualization, writing—original draft, writing—review and editing; S.L.H.: data curation, formal analysis, methodology, validation, writing—original draft, writing—review and editing.

All authors gave final approval for publication and agreed to be held accountable for the work performed therein.

Competing interests. We declare we have no competing interests.

Funding. We received no funding for this study.

Acknowledgements. The authors are grateful to Dr Andrew Bowgen, the NERC-funded PhD student who undertook this study as part of the British Antarctic Survey, Biological Responses to Extreme Antarctic Conditions and Hyper-extremes programme which was funded by NERC. In addition, the authors would like to thank the Rothera Research Station diving and boating teams. Lastly, the authors would also like to thank the anonymous reviewers whose comments have improved this manuscript.

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
