## [Peer Review File · Royal Society Open Science]

Review History

RSOS-211272.R0 (Original submission)

Review form: Reviewer 1

Is the manuscript scientifically sound in its present form?

Yes

Are the interpretations and conclusions justified by the results?

Yes

Is the language acceptable?

Yes

Do you have any ethical concerns with this paper?

Yes

Have you any concerns about statistical analyses in this paper?

No

Recommendation?

Accept with minor revision (please list in comments)

Comments to the Author(s)

Such a study is long overdue and it's great to see data from whole animal protein metabolism in an Antarctic teleost species at different acclimation temperatures. The study confirms previous predictions derived from a comparison of polar invertebrate species and temperate teleost species i.e. that there is reduced ability to synthesise proteins in the extreme cold. The strength here is the associated measurement of growth rates to enable an estimation of protein degradation rates. The study could be criticised for a two species comparison which seem to have been conducted in different locations at different times. However, the methodology is the same, the authors have standardised for food ration and body size, and these experiments are not easy because of the use of a radioisotope. The text is well written but lacks analysis in some places and would benefit from a bit more detail in others.

Specific comments

Consider changing 'polar' to 'Antarctic'. A study on a single species of Antarctic fish species is unlikely to be representative of all polar fish species.

Summary: I think that it is important to emphasise that this study is reporting on whole-animal rates of protein metabolism. Previous studies have reported on protein synthesis in individual tissues from polar teleosts. As this is comparison between 2 species, some justification is needed for the choice of the temperature species. The overall conclusion of this paper is not new because this response was predicted previously when comparisons were made among polar invertebrate and temperature teleosts. Overall, this study suggest more of a low temperature effect rather than variations based on phylogenetic differences.

Line 89 Understanding?

Materials and Methods

Missing information on the length of time *H. antarcticus* were held at 0 degrees C before temperature acclimation started. There's no reference to the location in which the fish were held and the experiments conducted.

Lines 108 and 120. There is discrepancy in the terminology used. Is this temperature acclimatisation or acclimation?

There is no mention of RNA:protein ratios and RNA activity in the main text of the paper. For clarity, it would be preferable to see a brief explanation of both parameters before the Results.

Results. Lines 172 and 188. If there are no statistical differences then it can be argued that temperature does not have an effect. Better to report as stated on line 310.

Conclusion

Line 303. This response may also be related to differences in food absorption.

Line 317. This response suggests differences in Q10 values depending on thermal histories, as previously reported in the literature.

Line 436. Further analysis would be good here. e.g. relationship to growth rates.

Lines 439-442. There is no analysis of the responses in RNA: protein and RNA translational efficiency. Some discussion of the elevation in RNA:protein observed in *H. antarcticus* is required.

Review form: Reviewer 2

Is the manuscript scientifically sound in its present form?

Yes

Are the interpretations and conclusions justified by the results?

Yes

Is the language acceptable?

Yes

Do you have any ethical concerns with this paper?

No

Have you any concerns about statistical analyses in this paper?

No

Recommendation?

Major revision is needed (please make suggestions in comments)

Comments to the Author(s)

General

This manuscript reports comparative work on the protein metabolism of an Antarctic and temperate fish. The authors measured temperature dependent whole animal growth, feeding rates and in vivo protein synthesis, degradation and protein synthesis retention efficiency. Based on the results the authors suggest that "polar fish degrade a significantly larger proportion of synthesised protein than temperate fish, with fundamental energetic implications for growth at low temperatures".

We congratulate the authors on this interesting study. In general, the ms is well written and the interpretation is good but we do have some comments that should be considered before publication.

Major comments:

1) In order to better evaluate the results and the temperature ranges chosen for the acclimation experiments (H. antarcticus: -1, 1 and 3°C, L. pholis: 3, 8, 13 and 18°C), it is important to obtain information about the habitat of the two fish species. Furthermore, the statement "ecologically similar fishes" should be elaborated for the studied species. In addition, it is important to know at what time of year the authors conducted the experiments. Possible side effects due to seasonality should be addressed/discussed in the discussion. See for example the study of Lewis & Driedzic (2007), *Am J Physiol Regul Integr Comp Physiol* 293: R474–R481, doi:10.1152/ajpregu.00594.2006.

2) Some information is missing in the M&M part, which makes it difficult to understand:

Please clarify when exactly the experiment was started (lines 120-124). Your sentence "The fish were weighed on the first and penultimate day of acclimatization" (line 123f) is misleading. You mention that "...Antarctic fish take 3-4 weeks to acclimate to higher temperatures" (lines 122-123). This implies that the animals were brought to the respective experimental temperature (lines 116-120) and were acclimated for 28 days at the respective temperature before the start of the experiment, i.e., the experiments started for another 28 days to determine growth, daily food intake, and protein metabolism in fully acclimated animals. If not, how can you exclude possible side effects when animals were still in the acclimation phase?

Please clarify when the fish were last fed prior to sampling and protein metabolism measurements. You discuss the aspect of SDA on protein synthesis (line 292ff), but this is not clear in the experimental protocol.

Finally, we strongly recommend that the information on measurement/extraction details from Suppl. M&M be included in the main text. In addition, no information is provided on the extraction and determination of RNA and kRNA. Please explain why you measured protein metabolism in whole animal extracts and not muscle tissue, since muscle protein metabolism is considered the most important mechanism for regulating growth intensity. I suspect one reason for this may be the small size of the fish species, but it would be good to see an explanation/discussion in the MS. Throughout the MS (if referring to literature data), please indicate whether whole-body ks or tissue-specific ks are meant, as well as vertebrates and invertebrates (e.g., line 320).

Minor comments:

Line 66: Consider also citing "17" as it is a comparative study as well.

Line 75-79: Please specify, tissue-dependent or whole animal levels? Against the background of protein metabolism and growth, it is worth noting that white muscle tissue (which accounts for ~50% of total fish body weight) has very low values. See e.g. Nemova et al. 2016, Russian Journal of Developmental Biology, 2016, Vol. 47, No. 4, pp. 161-172

Line 87-93: Please also include the hypotheses underlying the study.

Line 98f: Please add the common names of the species. Information on environmental parameters and timing of animal collection is lacking.

Line 104: How often was *L. pholis* fed?

Line 108: Please change acclimatisation into acclimation throughout the ms as acclimatisation refers to physiological adjustments in the field and not the lab (see Clarke 1991). Please comment on the use of "only" 3 temperatures for *H. antarcticus*, in particular after your statement "most studies...of either two, or a very limited range of temperatures..." (line 253f).

Line 109: Please include info of mean body weight and length here, in addition to T-dependent metrics in Table 1s. Are the fish considered juveniles or adults?

Line 111: Please define "clean".

Line 127: Please explain why you used the ln of M.

Results in general: Even if there is no significant relationship with temperature for some parameters, wouldn't it make sense to set up an equation anyway?

Line 191: We are not confident with the cubic model. With an equation (3rd order polynomial) you can describe any curve but is the relationship really biologically meaningful. Consider if it is useful to display the data as an Arrhenius plot.

Line 205f: We recommend that the equations be included in the figure legends and that Tables 1 and 2 be included in Suppl. Materials.

Line 275: The paragraph "Validation..." is rather descriptive than a real discussion. We agree with the authors that this aspect needs to be mentioned but perhaps it is better to mention it in the M&M or results section.

Line 286: Please define ks when first mentioned in the discussion.

Line 332: Please add "whole animal" before ks to make clear that your conclusion refers only to ks for the whole animal. This is because tissue specific ks values may be different.

Line 336: Please clarify what you mean with "at typical growth temperatures".

Line 364 and 430: Shouldn't it be 13°C? We recommend staying with the original data rather than using calculated ones due to the limited data sets and cubic model used. See comment to line 191.

Line 366: Please clarify in what respect the data are "broadly similar to" literature data.

Line 374: Please be more specific. E.g citation 65 determined whole animal growth rates (SGR) rather than kg. Furthermore, different size classes (10-60g vs. 100g vs. 3-5kg) were studied and size-dependent growth rates (%/day) varied from 0.2 to 2 at the respective optimum temperature.

Line 361f: The paragraph is descriptive and compares present with literature data but what is the message of it? Consider revising.

Line 389f: You mention the ubiquitin-proteasome pathway for protein degradation. However, in white muscle tissue (most relevant tissue for growth performance) this pathway is less relevant compared to others (see also comment to line 75f, and Nemova et al. 2016). This aspect should be considered/discussed. Just a question out of curiosity: Why didn't you additionally measure degradation pathways to better link your estimation of in vivo protein degradation?

Line 416: Please add more info if polar ectotherms have indeed elevated levels of oxidative damage and ROS compared to temperate ones. See for example Heise et al. 2007, *J Comp Physiol B*, doi: 10.1007/s00360-007-0173-4

Line 417f: Are there similar studies in temperate or tropical ectotherms?

Line 427f: Please add a little more detail for better understanding.

Line 431: This is speculative because there are no data between 13 and 18°C. Please revise sentence.

Line 436: "comparatively high" to what?

Line 433 – 442: Consider revising. See also comment to line 275.

Line 444ff: The conclusions are a little weak. Consider rewriting them.

Line 451f: Please comment on why A Bowgen is not among the authors.

Figures:

In general, legends lack details such as acclimation period etc. They should contain all the information and be understandable to the reader without reading the main text.

Please highlight significant differences between species at 3°C.

Please add N numbers.

Supplementary Material:

Please add how exactly fish was killed? Did you use the weight of the gutted fish? Please add the ratio of PCA:fish weight used. The statement "fish were homogenized in a known volume ...of PCA" is incorrect. Please add the info that washing the pellet twice was sufficient to remove unbound Phe.

Table 1s: Units are missing.

Chapter: Validation of the flooding dose:

Please add "methodology" in the chapter's head line

Both protein-bound and free-pool Phe increased by 3.7 or can it be a typo?

Please add the absolute values of Phe levels as well.

Please include information on whether baseline levels of unlabeled Phe were similar across species and how baseline levels were incorporated into the calculation.

Fig.1s: We recommend standardizing the axis units.

Please add formula/more info on how exactly protein degradation was calculated. It is difficult to understand for non-experts.

Decision letter (RSOS-211272.R0)

Dear Dr Fraser

The Editors assigned to your paper RSOS-211272 "Life in the freezer: protein metabolism in polar fish" have now received comments from reviewers and would like you to revise the paper in accordance with the reviewer comments and any comments from the Editors. Please note this decision does not guarantee eventual acceptance.

Please submit your revised manuscript and required files (see below) no later than 21 days from today's (ie 09-Nov-2021) date. Note: the ScholarOne system will 'lock' if submission of the revision is attempted 21 or more days after the deadline. If you do not think you will be able to meet this deadline please contact the editorial office immediately.

Thank you for submitting your interesting manuscript to Royal Society Open Science and we look forward to receiving your revision. If you have any questions at all, please do not hesitate to get in touch.

on behalf of Dr Ed Bolt (Associate Editor) and Malcolm White (Subject Editor)
openscience@royalsociety.org

Associate Editor Comments to Author (Dr Ed Bolt):

Associate Editor: 1

Comments to the Author:

Please submit a revised manuscript that takes account of the reviewer comments.

Reviewer comments to Author:

Reviewer: 1

Comments to the Author(s)

Such a study is long overdue and it's great to see data from whole animal protein metabolism in an Antarctic teleost species at different acclimation temperatures. The study confirms previous predictions derived from a comparison of polar invertebrate species and temperate teleost species i.e. that there is reduced ability to synthesise proteins in the extreme cold. The strength here is the associated measurement of growth rates to enable an estimation of protein degradation rates. The

study could be criticised for a two species comparison which seem to have been conducted in different locations at different times. However, the methodology is the same, the authors have standardised for food ration and body size, and these experiments are not easy because of the use of a radioisotope. The text is well written but lacks analysis in some places and would benefit from a bit more detail in others.

Specific comments

Consider changing 'polar' to 'Antarctic'. A study on a single species of Antarctic fish species is unlikely to be representative of all polar fish species.

Summary: I think that it is important to emphasise that this study is reporting on whole-animal rates of protein metabolism. Previous studies have reported on protein synthesis in individual tissues from polar teleosts. As this is comparison between 2 species, some justification is needed for the choice of the temperature species. The overall conclusion of this paper is not new because this response was predicted previously when comparisons were made among polar invertebrate and temperate teleosts. Overall, this study suggests more of a low temperature effect rather than variations based on phylogenetic differences.

Line 89 Understanding?

Materials and Methods

Missing information on the length of time *H. antarcticus* were held at 0 degrees C before temperature acclimation started. There's no reference to the location in which the fish were held and the experiments conducted.

Lines 108 and 120. There is discrepancy in the terminology used. Is this temperature acclimatisation or acclimation?

There is no mention of RNA:protein ratios and RNA activity in the main text of the paper. For clarity, it would be preferable to see a brief explanation of both parameters before the Results.

Results. Lines 172 and 188. If there are no statistical differences then it can be argued that temperature does not have an effect. Better to report as stated on line 310.

Conclusion

Line 303. This response may also be related to differences in food absorption.

Line 317. This response suggests differences in Q10 values depending on thermal histories, as previously reported in the literature.

Line 436. Further analysis would be good here. e.g. relationship to growth rates.

Lines 439-442. There is no analysis of the responses in RNA: protein and RNA translational efficiency. Some discussion of the elevation in RNA:protein observed in *H. antarcticus* is required.

Reviewer: 2

Comments to the Author(s)

General

This manuscript reports comparative work on the protein metabolism of an Antarctic and temperate fish. The authors measured temperature dependent whole animal growth, feeding rates and in vivo protein synthesis, degradation and protein synthesis retention efficiency. Based on the results the authors suggest that "polar fish degrade a significantly larger proportion of

synthesised protein than temperate fish, with fundamental energetic implications for growth at low temperatures”.

We congratulate the authors on this interesting study. In general, the ms is well written and the interpretation is good but we do have some comments that should be considered before publication.

Major comments:

1) In order to better evaluate the results and the temperature ranges chosen for the acclimation experiments (*H. antarcticus*: -1, 1 and 3°C, *L. pholis*: 3, 8, 13 and 18°C), it is important to obtain information about the habitat of the two fish species. Furthermore, the statement "ecologically similar fishes" should be elaborated for the studied species. In addition, it is important to know at what time of year the authors conducted the experiments. Possible side effects due to seasonality should be addressed/discussed in the discussion. See for example the study of Lewis & Driedzic (2007), *Am J Physiol Regul Integr Comp Physiol* 293: R474–R481, doi:10.1152/ajpregu.00594.2006.

2) Some information is missing in the M&M part, which makes it difficult to understand: Please clarify when exactly the experiment was started (lines 120-124). Your sentence "The fish were weighed on the first and penultimate day of acclimatization" (line 123f) is misleading. You mention that "...Antarctic fish take 3-4 weeks to acclimate to higher temperatures" (lines 122-123). This implies that the animals were brought to the respective experimental temperature (lines 116-120) and were acclimated for 28 days at the respective temperature before the start of the experiment, i.e., the experiments started for another 28 days to determine growth, daily food intake, and protein metabolism in fully acclimated animals. If not, how can you exclude possible side effects when animals were still in the acclimation phase?

Please clarify when the fish were last fed prior to sampling and protein metabolism measurements. You discuss the aspect of SDA on protein synthesis (line 292ff), but this is not clear in the experimental protocol.

Finally, we strongly recommend that the information on measurement/extraction details from Suppl. M&M be included in the main text. In addition, no information is provided on the extraction and determination of RNA and kRNA. Please explain why you measured protein metabolism in whole animal extracts and not muscle tissue, since muscle protein metabolism is considered the most important mechanism for regulating growth intensity. I suspect one reason for this may be the small size of the fish species, but it would be good to see an explanation/discussion in the MS. Throughout the MS (if referring to literature data), please indicate whether whole-body ks or tissue-specific ks are meant, as well as vertebrates and invertebrates (e.g., line 320).

Minor comments:

Line 66: Consider also citing "17" as it is a comparative study as well.

Line 75-79: Please specify, tissue-dependent or whole animal levels? Against the background of protein metabolism and growth, it is worth noting that white muscle tissue (which accounts for ~50% of total fish body weight) has very low values. See e.g. Nemova et al. 2016, *Russian Journal of Developmental Biology*, 2016, Vol. 47, No. 4, pp. 161-172

Line 87-93: Please also include the hypotheses underlying the study.

Line 98f: Please add the common names of the species. Information on environmental parameters and timing of animal collection is lacking.

Line 104: How often was *L. pholis* fed?

Line 108: Please change acclimatisation into acclimation throughout the ms as acclimatisation refers to physiological adjustments in the field and not the lab (see Clarke 1991). Please comment on the use of "only" 3 temperatures for *H. antarcticus*, in particular after your statement "most studies...of either two, or a very limited range of temperatures..." (line 253f).

Line 109: Please include info of mean body weight and length here, in addition to T-dependent metrics in Table 1s. Are the fish considered juveniles or adults?

Line 111: Please define “clean”.

Line 127: Please explain why you used the ln of M.

Results in general: Even if there is no significant relationship with temperature for some parameters, wouldn't it make sense to set up an equation anyway?

Line 191: We are not confident with the cubic model. With an equation (3rd order polynomial) you can describe any curve but is the relationship really biologically meaningful. Consider if it is useful to display the data as an Arrhenius plot.

Line 205f: We recommend that the equations be included in the figure legends and that Tables 1 and 2 be included in Suppl. Materials.

Line 275: The paragraph “Validation...” is rather descriptive than a real discussion. We agree with the authors that this aspect needs to be mentioned but perhaps it is better to mention it in the M&M or results section.

Line 286: Please define k_s when first mentioned in the discussion.

Line 332: Please add “whole animal” before k_s to make clear that your conclusion refers only to k_s for the whole animal. This is because tissue specific k_s values may be different.

Line 336: Please clarify what you mean with “at typical growth temperatures”.

Line 364 and 430: Shouldn't it be 13°C? We recommend staying with the original data rather than using calculated ones due to the limited data sets and cubic model used. See comment to line 191.

Line 366: Please clarify in what respect the data are “broadly similar to” literature data.

Line 374: Please be more specific. E.g citation 65 determined whole animal growth rates (SGR) rather than kg. Furthermore, different size classes (10-60g vs. 100g vs. 3-5kg) were studied and size-dependent growth rates (%/day) varied from 0.2 to 2 at the respective optimum temperature.

Line 361f: The paragraph is descriptive and compares present with literature data but what is the message of it? Consider revising.

Line 389f: You mention the ubiquitin-proteasome pathway for protein degradation. However, in white muscle tissue (most relevant tissue for growth performance) this pathway is less relevant compared to others (see also comment to line 75f, and Nemova et al. 2016). This aspect should be considered/discussed. Just a question out of curiosity: Why didn't you additionally measure degradation pathways to better link your estimation of in vivo protein degradation?

Line 416: Please add more info if polar ectotherms have indeed elevated levels of oxidative damage and ROS compared to temperate ones. See for example Heise et al. 2007, J Comp Physiol B, doi: 10.1007/s00360-007-0173-4

Line 417f: Are there similar studies in temperate or tropical ectotherms?

Line 427f: Please add a little more detail for better understanding.

Line 431: This is speculative because there are no data between 13 and 18°C. Please revise sentence.

Line 436: “comparatively high” to what?

Line 433 – 442: Consider revising. See also comment to line 275.

Line 444ff: The conclusions are a little weak. Consider rewriting them.

Line 451f: Please comment on why A Bowgen is not among the authors.

Figures:

In general, legends lack details such as acclimation period etc. They should contain all the information and be understandable to the reader without reading the main text.

Please highlight significant differences between species at 3°C.

Please add N numbers.

Supplementary Material:

Please add how exactly fish was killed? Did you use the weight of the gutted fish? Please add the ratio of PCA:fish weight used. The statement “fish were homogenized in a known volume ...of

PCA" is incorrect. Please add the info that washing the pellet twice was sufficient to remove unbound Phe.

Table 1s: Units are missing.

Chapter: Validation of the flooding dose:

Please add "methodology" in the chapter's head line

Both protein-bound and free-pool Phe increased by 3.7 or can it be a typo?

Please add the absolute values of Phe levels as well.

Please include information on whether baseline levels of unlabeled Phe were similar across species and how baseline levels were incorporated into the calculation.

Fig.1s: We recommend standardizing the axis units.

Please add formula/more info on how exactly protein degradation was calculated. It is difficult to understand for non-experts.

===PREPARING YOUR MANUSCRIPT===

If you have been asked to revise the written English in your submission as a condition of publication, you must do so, and you are expected to provide evidence that you have received language editing support. The journal would prefer that you use a professional language editing service and provide a certificate of editing, but a signed letter from a colleague who is a fluent speaker of English is acceptable. Note the journal has arranged a number of discounts for authors using professional language editing services (<https://royalsociety.org/journals/authors/benefits/language-editing/>).

===PREPARING YOUR REVISION IN SCHOLARONE===

Author's Response to Decision Letter for (RSOS-211272.R0)

See Appendices A & B.

Decision letter (RSOS-211272.R1)

Dear Dr Fraser,

It is a pleasure to accept your manuscript entitled "Life in the freezer: protein metabolism in Antarctic fish" in its current form for publication in Royal Society Open Science. The comments of the reviewer(s) who reviewed your manuscript are included at the foot of this letter.

on behalf of Dr Ed Bolt (Associate Editor) and Malcolm White (Subject Editor)
openscience@royalsociety.org

Associate Editor Comments to Author (Dr Ed Bolt):
Associate Editor

Comments to the Author:

Pleased that the revised manuscript can now be accepted - thorough responses and edits have been made in line with requests from both reviewers.

Appendix A

Authors responses to Referee 1

Author's responses in blue. All line numbers refer to track changed version

Reviewer comments to Author:

Reviewer: 1

Comments to the Author(s)

Such a study is long overdue and it's great to see data from whole animal protein metabolism in an Antarctic teleost species at different acclimation temperatures. The study confirms previous predictions derived from a comparison of polar invertebrate species and temperate teleost species i.e. that there is reduced ability to synthesise proteins in the extreme cold. The strength here is the associated measurement of growth rates to enable an estimation of protein degradation rates. The study could be criticised for a two species comparison which seem to have been conducted in different locations at different times. It wasn't and clarification has been added in lines 117-121. However, the methodology is the same, the authors have standardised for food ration and body size, and these experiments are not easy because of the use of a radioisotope. The text is well written but lacks analysis in some places and would benefit from a bit more detail in others.

Specific comments

Consider changing 'polar' to 'Antarctic'. A study on a single species of Antarctic fish species is unlikely to be representative of all polar fish species. Change made throughout manuscript

Summary: I think that it is important to emphasise that this study is reporting on whole-animal rates of protein metabolism. Previous studies have reported on protein synthesis in individual tissues from polar teleosts. As this is comparison between 2 species, some justification is needed for the choice of the temperature species. The overall conclusion of this paper is not new because this response was predicted previously when comparisons were made among polar invertebrate and temperature teleosts. Overall, this study suggests more of a low temperature effect rather than variations based on phylogenetic differences. Have now increased emphasis that the study is reporting on whole-animal rather than tissue protein synthesis rates throughout. Further details have been added in lines 121-127, regarding ecological details of the species used and why the temperate species was selected.

Line 89 Understanding? Corrected, see line 92.

Materials and Methods

Missing information on the length of time *H. antarcticus* were held at 0 degrees C before temperature acclimation started. Added information on line 112. There's no reference to the location in which the fish were held and the experiments conducted. Further details added 117-121.

Lines 108 and 120. There is discrepancy in the terminology used. Is this temperature acclimatisation or acclimation? Changed to temperature acclimation throughout manuscript

There is no mention of RNA:protein ratios and RNA activity in the main text of the paper. For clarity, it would be preferable to see a brief explanation of both parameters before the Results. Apologies this information must have been accidentally removed in an edit. Information added on lines 95

(Introduction), 182-185 and 217-222 (Material and Methods).

Results. Lines 172 and 188. If there are no statistical differences then it can be argued that temperature does not have an effect. Better to report as stated on line 310. Changed as per the referee's suggestions, see lines 239-241 & 267.

Conclusion

Line 303. This response may also be related to differences in food absorption. The authors agree and some text has been added to reflect this point see lines 390-91

Line 317. This response suggests differences in Q10 values depending on thermal histories, as previously reported in the literature. Section added 406-410 to make this point.

Line 436. Further analysis would be good here. e.g. relationship to growth rates.

The authors have previously considered carrying out an analysis of the relationship between PSRE and growth rates (SGR) but we would be concerned about the independence of the variables for the analysis. PSRE is calculated from protein growth (k_g)/ protein synthesis (k_s)*100 and k_g is calculated from the initial and final animal protein contents using the Ricker growth equation that is also used for calculating SGR. As in this study, protein content of the animals varied little between the beginning and end of the experiment the relationship between SGR and protein growth is very close (see Table 2 equations for *L. pholis*). So if we analyse the relation between SGR & PSRE both variables will include a component that has been calculated using the Ricker growth equation and in the case of this study where the protein content is very consistent over the course of the experimental period, we would have concerns regarding the independence of the two variables. Therefore we would not be keen to include this analysis.

Lines 439-442. There is no analysis of the responses in RNA: protein and RNA translational efficiency. Some discussion of the elevation in RNA:protein observed in *H. antarcticus* is required. Section added to analyse the RNA: Protein and RNA translational efficiency data and discuss the elevation of RNA concentrations seen in the Antarctic species 567-575.

Appendix B

Reviewer: 2

Authors responses to Referee 1

Author's responses in blue. All line numbers refer to track changed version

Comments to the Author(s)

General

This manuscript reports comparative work on the protein metabolism of an Antarctic and temperate fish. The authors measured temperature dependent whole animal growth, feeding rates and in vivo protein synthesis, degradation and protein synthesis retention efficiency. Based on the results the authors suggest that "polar fish degrade a significantly larger proportion of synthesised protein than temperate fish, with fundamental energetic implications for growth at low temperatures".

We congratulate the authors on this interesting study. In general, the ms is well written and the interpretation is good but we do have some comments that should be considered before publication.

Major comments:

1) In order to better evaluate the results and the temperature ranges chosen for the acclimation experiments (H. antarcticus: -1, 1 and 3°C, L. pholis: 3, 8, 13 and 18°C), it is important to obtain information about the habitat of the two fish species Text added see lines 123-127. Furthermore, the statement "ecologically similar fishes" should be elaborated for the studied species (Text added see. Text added see lines 121-123). In addition, it is important to know at what time of year the authors conducted the experiments. Text added see lines 118-121. Possible side effects due to seasonality should be addressed/discussed in the discussion. See for example the study of Lewis & Driedzic (2007), Am J Physiol Regul Integr Comp Physiol 293: R474–R481, doi:10.1152/ajpregu.00594.2006. Text added to discuss this point - lines 430-438.

2) Some information is missing in the M&M part, which makes it difficult to understand: Please clarify when exactly the experiment was started (lines 120-124). Your sentence "The fish were weighed on the first and penultimate day of acclimatization" (line 123f) is misleading. Extra text added to clarify this point, see lines 143-144 and 148. You mention that "...Antarctic fish take 3-4 weeks to acclimate to higher temperatures" (lines 122-123). This implies that the animals were brought to the respective experimental temperature (lines 116-120) and were acclimated for 28 days at the respective temperature before the start of the experiment, i.e., the experiments started for another 28 days to determine growth, daily food intake, and protein metabolism in fully acclimated animals. This is correct, the animals were gradually brought to their respective experimental temperatures and when at that that temperature acclimated for 28 days whilst growth and food consumption were measured and protein synthesis was measured at the end of this period. Information added at lines 143-144 and 148 now clarify this point. If not, how can you exclude possible side effects when animals were still in the acclimation phase?

Please clarify when the fish were last fed prior to sampling and protein metabolism measurements. You discuss the aspect of SDA on protein synthesis (line 292ff), but this is not clear in the experimental protocol. Text added at 159-160 to clarify this point.

Finally, we strongly recommend that the information on measurement/extraction details from Suppl. M&M be included in the main text. This information has now been moved from the supplementary material into the manuscript see lines 117-195. As the methods are quite well established and have been published numerous times, the authors had thought this would shorten the overall manuscript, but we are happy to include the method in the manuscript. In addition, no information is provided on the extraction and determination of RNA and kRNA. A section has now been added to the methods with this information see lines 182-185. Please explain why you measured protein metabolism in whole animal extracts and not muscle tissue, since muscle protein metabolism is considered the most important mechanism for regulating growth intensity. I suspect one reason for this may be the small size of the fish species, but it would be good to see an explanation/discussion in the MS The whole point of the study was to be able to relate whole animal growth, food consumption and protein degradation, if we had only measured protein synthesis in white muscle this would have been impossible. Growth of muscle is undoubtedly important but if you want to measure whole animal growth then you need to measure that, white muscle won't suffice, lines added at 97-99 to clarify this decision. Throughout the MS (if referring to literature data), please indicate whether whole-body ks or tissue-specific ks are meant, as well as vertebrates and invertebrates (e.g., line 320). This has been clarified throughout the manuscript.

Minor comments:

Line 66: Consider also citing "17" as it is a comparative study as well. Citation added see line 68.

Line 75-79: Please specify, tissue-dependent or whole animal levels? This has been clarified throughout the manuscript. Against the background of protein metabolism and growth, it is worth noting that white muscle tissue (which accounts for ~ 50% of total fish body weight) has very low values. See e.g. Nemova et al. 2016, Russian Journal of Developmental Biology, 2016, Vol. 47, No. 4, pp. 161–172 Noted, we have now made this point at lines 507-512.

Line 87-93: Please also include the hypotheses underlying the study. Hypothesis now added at the end of the Introduction lines 99-101.

Line 98f: Please add the common names of the species. Common names of both species added at lines 106, 113. Information on environmental parameters and timing of animal collection is lacking Text added see lines 108, 123-127. The authors do not know the exact time of collection of the *L. pholis* as they were provided by a commercial supplier.

Line 104: How often was *L. pholis* fed? Text to clarify this point added on line 115.

Line 108: Please change acclimatisation into acclimation throughout the ms as acclimatisation refers to physiological adjustments in the field and not the lab (see Clarke 1991). Changed throughout the manuscript. Please comment on the use of "only" 3 temperatures for *H. antarcticus*, in particular after your statement "most studies...of either two, or a very limited range of temperatures..." (line 253f). *Harpagifer antarcticus* is only found in Antarctica and lives within a thermal regime that never exceeds 3°C. This species will not tolerate long-term acclimation to higher temperatures and hence it was not possible to feasibly include more than 3 temperatures within the temperate range the species will tolerate. Home Office licensing requirements would not allow us to expose these animals to temperatures that would potentially result in mortalities. *Lipophrys pholis* is fairly eurythermal and exists from Norway to Portugal and can tolerate a wide range of temperatures, *H. antarcticus* is very stenothermal and can't, hence the experimental temperatures ranges used.

Line 109: Please include info of mean body weight and length here, in addition to T-dependent metrics in Table 1s. Mean weights have been added on line 132. We did not record fish lengths as they are very difficult to measure accurately in fish this small and with this body shape. The aim was to measure mass as quickly as possible to minimise stress to the animals and we didn't need lengths. Are the fish considered juveniles or adults? All animals were considered adults and this has been clarified in line 127

Line 111: Please define "clean". Clean has just been removed as it was superfluous

Line 127: Please explain why you used the Ln of M. The equation used for specific growth rate is the standard Ricker growth equation from 1979 and this uses Ln Mass. This is a standard equation in fish growth.

Results in general: Even if there is no significant relationship with temperature for some parameters, wouldn't it make sense to set up an equation anyway? The authors would strongly argue against this suggestion, we have indicated in the text where there was a trend in a relationship between variables that was evident, but did not reach significant levels. However, we do not think it would be appropriate at all, to include equations describing non-significant relationships. Implicit in including the equation would be that there is a functional relationship between the variables when we have not statistically demonstrated one.

Line 191: We are not confident with the cubic model. With an equation (3rd order polynomial) you can describe any curve but is the relationship really biologically meaningful. We did in fact trial fit linear, quadratic and cubic models to the data sets (Fig 2CD, *L. pholis*) that the referee is talking about. Neither, linear, or quadratic models fit at all well, as the species has passed its growth optima and protein degradation rapidly increases and PSRE rapidly decreases. We appreciate that with a 3rd order polynomial curve it can describe any curve, but in this case we would argue that in fact it is biologically meaningful and the best way of describing the data, the data is not well represented by a quadratic model. If the Editor insists we fit a quadratic model to the data we would do so, but would not consider this is the best way to represent the data. Consider if it is useful to display the data as an Arrhenius plot. The authors did consider the use of Arrhenius plots for the data but after discussions considered it best to present the data in the form that we have to make direct comparison with the majority of published data easier.

Line 205f: We recommend that the equations be included in the figure legends and that Tables 1 and 2 be included in Suppl. Materials. The authors strongly disagree from a stylistic perspective that this is a sensible route to go. For example we would have to include 8 equations and r^2 values in the figure legend for Fig 2 and then repeat the equations again in the supplementary data. This seems pointless and the figure legend will be huge. Also if we transfer Table 2 to the Supplementary Materials then the reader won't be able to see any probabilities without going online and accessing the Supp materials. Unless the Editor disagrees we would like to keep Tables 1 & 2 as they are.

Line 275: The paragraph "Validation..." is rather descriptive than a real discussion. We agree with the authors that this aspect needs to be mentioned but perhaps it is better to mention it in the M&M or results section. This section has now been moved to the Results lines 251-259

Line 286: Please define k_s when first mentioned in the discussion. k_s is now defined on line 373

Line 332: Please add "whole animal" before k_s to make clear that your conclusion refers only to k_s for the whole animal. This is because tissue specific k_s values may be different. Added see line 425

Line 336: Please clarify what you mean with "at typical growth temperatures". Clarification has been provided on what is meant by typical growth temperatures on line 442

Line 364 and 430: Shouldn't it be 13°C? Typo corrected, see line 470. We recommend staying with the original data rather than using calculated ones due to the limited data sets and cubic model

used. See comment to line 191. The k_g data was in fact correct for the 13°C experimental temperature but the temperature was incorrect in the manuscript, now corrected see line 471 and 555

Line 366: Please clarify in what respect the data are “broadly similar to” literature data. Clarification of what the authors meant is added to line 473

Line 374: Please be more specific. E.g citation 65 determined whole animal growth rates (SGR) rather than kg. Furthermore, different size classes (10-60g vs. 100g vs. 3-5kg) were studied and size-dependent growth rates (%/day) varied from 0.2 to 2 at the respective optimum temperature.

Citation 65 has now been removed as it did not directly refer to protein growth

Line 361f: The paragraph is descriptive and compares present with literature data but what is the message of it? Consider revising. Paragraph has been revised and includes a summary sentence, line 486-489.

Line 389f: You mention the ubiquitin-proteasome pathway for protein degradation. However, in white muscle tissue (most relevant tissue for growth performance) this pathway is less relevant compared to others (see also comment to line 75f, and Nemova et al. 2016). This aspect should be considered/discussed. Text has been added at Lines 507-512 to discuss the point the referee raises

Just a question out of curiosity: Why didn't you additionally measure degradation pathways to better link your estimation of in vivo protein degradation? We would have liked to directly measure rates of protein degradation via specific pathways but did not have the methods available in our lab. This is an area we are keen to explore going forward though. Although in the context of the current study it would not be possible to directly measure protein degradation in a way that would allow direct comparison with the fractional rates of protein synthesis, growth and degradation reported in this study. However, a comparative study of protein degradation via the key pathways in *L. pholis* and *H. antarcticus* would greatly aid understanding of temperature effects on protein degradation.

Line 416: Please add more info if polar ectotherms have indeed elevated levels of oxidative damage and ROS compared to temperate ones. See for example Heise et al. 2007, J Comp Physiol B, doi: 10.1007/s00360-007-0173-4. Extra information has been added to clarify and expand on this point at lines 539-544

Line 417f: Are there similar studies in temperate or tropical ectotherms? Clarification has been provided that no similar studies have been carried out in ectotherms generally, and not just polar ectotherms see lines 537-538.

Line 427f: Please add a little more detail for better understanding. Further detail added lines 552-553

Line 431: This is speculative because there are no data between 13 and 18°C. Please revise sentence. This sentence has been revised and the PSRE maxima has been quoted at the experimental temperature of 13°C, not the interpolated maximum temperature as per the model see line 556.

Line 436: “comparatively high” to what? Text added on line 562 to clarify this point.

Line 433 – 442: Consider revising. See also comment to line 275. This section has now been pretty much rewritten see lines 559-577.

Line 444ff: The conclusions are a little weak. Consider rewriting them. Conclusions rewritten see 580-590.

Line 451f: Please comment on why A Bowgen is not among the authors. Andy undertook his PhD a few years ago and is no longer working in science, although he is very pleased to see the data published he is extremely busy in his current role and wasn't interested in being involved in writing the manuscript. Hence he asked not to be included as an author and just added to the acknowledgements.

Figures:

In general, legends lack details such as acclimation period etc. They should contain all the information and be understandable to the reader without reading the main text. Further detail added to figure legends.

Please highlight significant differences between species at 3°C. Added to Fig 1 & 2

Please add N numbers. Added to Fig 1 and 2 figure legends.

Supplementary Material:

Please add how exactly fish was killed? Further detail added, lines 11-13 in supplementary materials.

Did you use the weight of the gutted fish? No we did not. If we had used the weight of the gutted fish then we would have not been measuring whole animal metrics. Please add the ratio of PCA:fish weight used. Ratio added to line 175 of the main manuscript (note this referee asked this section of methods to be moved from supplementary information, to the main paper). The statement “fish were homogenized in a known volume ...of PCA” is incorrect. (see authors previous comment) Please add the info that washing the pellet twice was sufficient to remove unbound Phe. Text added to lines 179-180 of the main manuscript to clarify this point.

Table 1s: Units are missing. Units added.

Chapter: Validation of the flooding dose:

Please add “methodology” in the chapter’s head line Added on line 25 of supplementary material

Both protein-bound and free-pool Phe increased by 3.7 or can it be a typo? Yes it was a typo, text added to clarify that we were only referring to free-pools on line 34 of supplementary materials.

Please add the absolute values of Phe levels as well. Line 56-57 supplementary materials, values added.

Please include information on whether baseline levels of unlabeled Phe were similar across species and how baseline levels were incorporated into the calculation. Unfortunately the PhD student who undertook the work is living in Canada and the raw data regarding species specific baseline Phe concentrations is in storage in the UK and not available so we can’t answer this question. The baseline Phe levels are not required in the fractional protein synthesis calculation (see equation line 200). The only reason that the baseline Phe levels are measured is to ensure that flooding levels are adequate.

Fig.1s: We recommend standardizing the axis units. Not really sure what the referee is meaning here, if we standardise the axis units on the free-pool and bound protein figs they will be unreadable. The bound protein values range between about 0.1 and 1.4 dpm.nmole Phe and the free-pool values between 800 and 1000 dpm.nmole Phe. This is obviously a 3-4 magnitude difference in values? If they are plotted at the same axis scales the bound data will be unreadable? The axis units are already the same in Fig 1s, dpm.nmole Phe? What we have plotted is the standard way of plotting this validation data.

Please add formula/more info on how exactly protein degradation was calculated. It is difficult to understand for non-experts. Text has been added to the supplementary materials online 67 to clarify how protein degradation was calculated.